# Novel PEGylated Liposomes Enhance Immunostimulating Activity of isRNA

**DOI:** 10.3390/molecules23123101

**Published:** 2018-11-27

**Authors:** Tatyana Kabilova, Elena Shmendel, Daniil Gladkikh, Nina Morozova, Mikhail Maslov, Elena Chernolovskaya, Valentin Vlassov, Marina Zenkova

**Affiliations:** 1Institute of Chemical Biology and Fundamental Medicine SB RAS, Lavrentieva ave. 8, Novosibirsk 630090, Russia; kabilova@niboch.nsc.ru (T.K.); medulla35@gmail.com (D.G.); elena_ch@niboch.nsc.ru (E.C.); vvlassov@mail.ru (V.V.); 2Institute of Fine Chemical Technologies, Moscow Technological University, Vernadskogo ave. 86, Moscow 119571, Russia; shmelka_87@mail.ru (E.S.); ngmoroz@mail.ru (N.M.); mamaslov@mail.ru (M.M.)

**Keywords:** cationic liposomes, PEG-containing lipoconjugates, transfection, non-viral delivery system, immunostimulating RNA, biodistribution

## Abstract

The performance of cationic liposomes for delivery of therapeutic nucleic acids in vivo can be improved and specifically tailored to certain types of cargo and target cells by incorporation of PEG-containing lipoconjugates in the cationic liposome’s composition. Here, we report on the synthesis of novel PEG-containing lipoconjugates with molecular masses of PEG 800, 1500 and 2000 Da. PEG-containing lipoconjugates were used as one of the components in liposome preparation with the polycationic amphiphile 1,26-bis(cholest-5-en-3β-yloxycarbonylamino)-7,11,16,20-tetra-azahexacosan tetrahydrochloride (2X3) and the lipid-helper dioleoylphosphatidylethanolamine (DOPE). We demonstrate that increasing the length of the PEG chain reduces the transfection activity of liposomes in vitro, but improves the biodistribution, increases the circulation time in the bloodstream and enhances the interferon-inducing activity of immunostimulating RNA in vivo.

## 1. Introduction

At present, the prospects of therapeutic preparations based on nucleic acids have been proven and small interfering RNA, microRNA, antisense oligonucleotides, aptamers and genomic editing systems are widely used in experimental biomedicine. The use of these biologically active substances in therapy is hampered by the fact that their effective targeted delivery to cells has not been solved to date. Two main approaches are used to ensure the delivery of therapeutic nucleic acids to target cells: the formation of complexes with different particles and polymers [1] and bioconjugation with lipophilic molecules, antibodies, aptamers and amphiphiles [2]. Unfortunately, currently used polymers are quite toxic, and the effectiveness of bioconjugates is manifested only at high concentrations due to their removal from the bloodstream through filtration in the kidneys and deposition of the drug trapped in endosomes within the cells followed by their slow release. Compositions containing cationic lipids that efficiently interact with a negatively charged nucleic acid to form lipoplexes are widely used for the delivery of therapeutic nucleic acids. It has been shown that cationic lipoplexes predominantly accumulate in highly vascularized organs: liver, kidney, heart and lungs, but they do not have any cell or tissue specificity. Therefore, lipoplexes carrying ligands to specific cellular receptors are used to develop delivery vehicles for therapeutic nucleic acids [3,4,5], which reduces the overall toxic effect on the body and ensures the accumulation of an effective therapeutic dose in the target organ. Cationic lipoplexes, which are highly effective for delivering nucleic acids in cell culture, are less effective in the body due to the toxicity associated with the interactions of a positively charged particle with proteins and blood cells, and also due to the recognition and absorption of these particles by macrophages [6]. Therefore, the main tasks of creating drugs based on nucleic acids are stabilization of the drug in the bloodstream, ensuring its primary accumulation in target cells, reducing the rate of excretion of the drug and improving bioavailability. It is important to provide a particle size of 50–100 nm for use in microvesicles in vivo; it is preferable that their surface be neutral or slightly negatively charged. Since the packing of nucleic acids into neutral liposomes is a difficult task, the development of therapeutic nucleic acids delivery carriers utilizes a variety of multilayer structures that shield the charge of the nucleic acids. Modification of the surface of liposomes, creating a protective layer, is considered an effective strategy for increasing the time of the drug in the bloodstream. Most often, polyethylene glycol (PEG) is used for this purpose, since PEGylated nanoparticles are stable in the bloodstream and are effectively delivered to tumor tissue due to the Enhanced permeability and Retention (EPR) effect. On the other hand, PEGylation impairs a number of properties of lipid nanoparticles, which are critical for achieving a therapeutic effect. Negative effects are connected with the fact that PEGylation impairs transfection properties of liposomes [7], hinders the release of the therapeutic cargo from endosomes [8] and in some cases causes accelerated blood clearance (ABS), and the contribution of these effects in pharmacokinetics depends on the size of the PEG, the type of lipoconjugate and the composition of the liposome [9,10,11]. 

Previously, we developed an effective liposomal nucleic acid delivery system based on the polycationic amphiphile 1,26-bis(cholest-5-en-3β-yloxycarbonylamino)-7,11,16,20-tetraazahexacosan tetrahydrochloride (2X3) and the lipid-helper dioleoylphosphatidylethanolamine (DOPE) [12,13,14], which effectively delivers nucleic acids in vitro and in vivo and showed that the incorporation of a folic acid-containing lipoconjugate into the lipid composition provides targeted delivery to tumors that express folate receptors at a high level [5]. 

Within this study, we synthesized PEG-containing lipoconjugates that differed in the molecular weight of PEG (800, 1500 and 2000 Da), examined the physicochemical properties of liposomes containing them and determined their ability to transfect plasmid DNA and siRNA in cell culture, as well as biodistribution and biological activity of the obtained lipoplexes in mice. It was determined that increasing the length of the PEG chain reduces the transfection activity of liposomes in vitro, but improving biodistribution increases the circulation time in the bloodstream and enhances the interferon-inducing activity of immunostimulating RNA (isRNA) in vivo.

## 2. Results

### 2.1. Synthesis of PEG-Containing Lipoconjugates, Preparation and Characterization of Liposomes 

We synthesized a series of PEG-containing lipoconjugates, **3a**–**c**, with PEG spacers of different lengths: from 18 to 52 ethylene glycol units (Figure 1A). The starting hydrophobic molecule used for the synthesis of PEG-containing lipoconjugates **3a**–**c** was *rac*-1-*О*-(4-nitrophenyloxycarbonyl)-2,3-di-*O*-tetradecylglycerol (**1**), which was treated with an excess of diamines (*O*,*O*′-bis(2-aminoethyl)-octadecaethylene glycol (MW ~800 Da), bis(3-aminopropyl)polyethylene glycol (MW ~1500 Da) and bis(amino)polyethylene glycol (MW ~2000 Da) to give mono-amino derivatives **2a**–**c** containing PEG spacers [5,15]. The free terminal amino group of compounds **2a**–**c** was protected by acetylation to yield the lipoconjugates **3a**–**c** (Figure 1A). The resulting products **3a**–**c** were isolated by ion-exchange column chromatography. The structure and purity of the PEG lipoconjugate was confirmed by NMR (^1^H and ^13^C) and mass spectrometry (see Materials and methods). 

Based on polycationic amphiphile 2X3, neutral phospholipid DOPE and one of the lipoconjugates **3a**–**c** (48:48:2 molar ratio), the liposomal compositions **P800**, **P1500** and **P2000** were prepared by the hydrating lipid thin-film method, followed by sonication as previously described [12]. PEGylated liposomes formed small compact particles in the solution with diameters from 50.6 ± 12.7 nm for **P2000** to 76.5 ± 13.5 nm for **P1500**, which is close to that of conventional liposome **L** (2X3:DOPE, 1:1 molar ratio) (95.7 ± 19.5 nm). The polydispersity index (PI) values indicated that the presence of PEG lipoconjugate does not influence the homogeneity of the liposomes: liposomes had similar PI values that varied from 0.207 ± 0.015 for **P800** to 0.260 ± 0.055 for **L** (Table 1). These data are in agreement with our recent results [5] showing that the addition of up to 2 M% of folate-based lipoconjugate to liposomal composition did not alter particle size or homogeneity. Liposomes F, containing lipoconjugate in which folate was attached via octadecaethylene glycol spacer, formed particles with a diameter of 60.7 ± 21.7 and a PI of 0.200 ± 0.050, similar to those for **P800**.

The cytotoxicity of cationic liposomes was evaluated by MTT test [16] using HEK 293 cells, wherefore the cells were incubated with the PEGylated or conventional liposomes at concentrations ranging from 1–80 μM. The IC_50_ values for PEGylated and conventional liposomes were not achieved at concentration 80 µM, the number of living cells in the population after incubation with 80 µM liposomes is shown in Table 1. It is worth mentioning that working concentration of liposomes for in vitro delivery of NA did not exceed 9 μM. These results are in agreement with our recent data [5,12], showing that very low toxicity of conventional liposome L was not altered by incorporation in liposomal formulation of 2 M% folate lipoconjugate.

### 2.2. Effect of PEGylation on Lipoplex Formation and siRNA Silencing Activity in Transgenic BHK IR780 Cells

Lipoplexes composed of siRNA with scrambled sequence (siScr) and cationic liposomes were prepared at N/P ratios 1/1, 2/1 and 4/1, and characterized by dynamic light scattering. The results showed (Table 2) that an increase in the amount of conventional or PEGylated liposomes in lipoplexes (from N/P = 1/1 to 4/1) corresponded to a decrease of particle size and polydispersity index and resulted in the formation of more compact particles ranging in size from 93.1 ± 4.1 nm for **P1500** to 206.0 ± 10.2 nm for **L** liposomes. Both size and homogeneity of PEGylated liposomes were smaller than those of liposomes **L**. It should be noted that the particles formed by NA and liposomes were larger and less homogeneous than liposomes alone. However, the size of PEGylated lipoplexes formed at N/P ≥ 2/1 did not exceed ~200 nm and PI did not exceed 0.35.

Large particles with both positive and negative ξ-potentials are formed at the ratio N/P = 1/1 (Table 2). The ξ-potential of lipoplexes becomes positive at a ratio of N/P ≥ 2/1. It is important to note that an increase in the length of PEG leads to a decrease in the ξ potential of the complexes, which may be due to the screening of the surface charge by PEG residues.

Previously, we have shown that liposomes **L** exhibited exceptionally high transfection efficiency (TE) both in the presence of serum and in serum-free conditions, providing efficient siRNA delivery and performance [12]. To study anti-EGFP siRNA (siEGFP) delivery by PEGylated liposomes we used transgenic BHK IR780 cells stably expressing EGFP. The reduction of EGFP-associated cell fluorescence after cell incubation with siEGFP/liposome complexes characterizes TE of liposomes in terms of cellular accumulation and bioperformance of delivered siRNA. As it was shown previously [12], the N/P ratios for siEGFP delivery by liposome **L** providing efficient gene silencing were 6/1 and 8/1. Here we used siEGFP/PEGylated liposomes at N/P from 4/1 to 8/1. Lipofectamine 2000 and liposomes **L** were used as positive controls.

The most efficient inhibition of EGFP expression was observed after siEGFP delivery by conventional **L** and PEGylated liposome **P800** containing lipoconjugate **3a** with the shortest PEG spacer shown (Figure 2). The TE of **P800** was similar (79–90% EGFP inhibition) at all N/P ratios used, both in serum-free conditions (Figure 2A, triangles) and in the presence of serum (Figure 2B, triangles). The TE of liposomes **L** was as high as **P800** under serum-free conditions and somewhat lower (65 and 80% EGFP inhibition, for **L** and **P800**, respectively) in the presence of serum only at N/P of 4/1. The TE of both liposomes **L** and **P800** became similar to that observed in serum-free conditions (inhibition of 84–88%) when N/P increased to 6/1 and 8/1. The TE of liposomes **P1500** and **P2000** was significantly lower and strongly depended on N/P ratio. At high N/P ratio (6/1, 8/1) the EGFP inhibition reached 30–35% in the absence and was less than 50% in the presence of serum (Figure 2A,B, dark circles, open squares). Lipofectamine 2000 led to inhibition of EGFP levels by 78% under serum-free conditions and by 65% in the presence of serum. Thus, based on these results we can conclude that addition of PEGylated lipoconjugated to liposomal composition **L** does not improve TE of these liposomes: formulation **P800** displayed similar properties as **L**; further increase of PEG length within the lipoconjugates had a negative effect on TE of these formulations in vitro. It is worth mentioning that formulation **P800** was more potent in the presence of serum, since even at N/P of 4/1 it provided more efficient EGFP silencing than formulation **L** and Lipofectamine2000 (both 65%).

### 2.3. Plasmid DNA Delivery into HEK 293 Cells

A similar set of experiments was performed to characterize the delivery of plasmid DNA (pEGFP-C2) mediated by liposomes **P800**, **P1500**, **P2000** and **L**. In this case TE was estimated by measuring the level of EGFP encoded by the plasmid [17]. As in the case of siRNA, the lipoplexes of pEGFP-C2 and liposomes were formed at N/P ratios of 4/1, 6/1 and 8/1. The results showed (Figure 3) that similar to siRNA, liposomes **P800** and **L**, as well as Lipofectamine 2000 efficiently deliver plasmid DNA into more than 60% of cells (Figure 3A) with mean fluorescence intensity (MFI) of 20–30 RFU (Figure 3B). Liposomes **P1500** and **P2000** were less effective both in terms of the number of EGFP-positive cells in the population (44–51% and 32–42% of cells) and mean fluorescent intensity (4.0–7.5 and ~1 RFU, for **P1500** and **P2000**, respectively). Thus, just as in the case of siEGFP delivery, liposomes **P800** and **L** exhibited remarkably high levels of pDNA delivery, while TE of liposomes with long PEG spacers, **P1500** and **P2000**, was significantly reduced especially in terms of the level of transgene expression within the cells.

### 2.4. Biodistribution of siRNA Complexed with PEGylated Liposomes in Mice

The biodistribution of siRNA complexed with PEGylated **P** or conventional **L** liposomes in SCID mice was investigated using Cy5.5-labeled siRNA (siMDR) targeted to human *MDR1* mRNA, which was no mRNA target in mice. Groups of three mice each were i.v. injected with lipoplexes siMDR/liposomes formed at N/P = 4/1. A non-injected mouse was used as a control. Lifetime multispectral fluorescent imaging analysis was performed to evaluate the dynamics of Cy5.5 labeled siMDR biodistribution in the mouse body (for details see Materials and methods). Mice from different experimental groups were injected and imaged simultaneously at the specified time points. Figure 4 shows representative images of mice taken at 20 min and 24 h post injection (p.i.).

The results showed that lipoplexes siMDR/**L** and siMDR/**P** rapidly spread throughout the mouse in the bloodstream, and 20 min p.i. the fluorescence signal was detected in the whole body (Figure 4A). The fluorescence signal decreased with time, and 24 h p.i. was concentrated in the areas of internal organs (Figure 4B). The efficiency of accumulation of fluorescence signals depends on the presence of PEG in liposomes: biodistribution of lipoplexes containing **P800**, **P1500** or **P2000** was characterized by higher retention in the mouse 24 h p.i. in comparison with siMDR/**L**.

After 24 h, the internal organs of the mouse were extracted and the total amount of fluorescence accumulated in the organs was calculated as total fluorescence from the heart, lungs, liver, spleen and kidneys, and varied from 300 to 550 RFU (Figure 4C, Table 3). The results reveal (Figure 4C, Table 3) that siMDR complexed with PEGylated **P800**, **P1500** or conventional **L** liposomes accumulated mainly in the following organs: kidneys (~30%), lungs (~28%), liver (~25%), spleen (~15%) and to a lesser extent in the heart (~2.5%). An augmentation in the length of PEG-spacer leads to an increase in the accumulation of siMDR/**P2000** in the kidneys (40.0%) and heart (10.5%), and to a decrease in the accumulation in the liver (15.5%) and spleen (7.5%). 

Thus, lipoconjugates containing PEG with a mol. weight of 800–1500 kDa can be introduced into liposomes up to an amount of 2% without reducing the efficiency of their accumulation in the mouse organs, the use of a lipoconjugate with a longer PEG can be used to increase the total accumulation of the delivered siRNA, and especially accumulation in heart tissue. 

### 2.5. Dynamics of siRNA Concentration in Mouse Blood Plasma after Intravenous Administration of its Complexes with P and L Liposomes

PEG-containing liposomes loaded with low molecular weight drugs demonstrate improved pharmacokinetic properties and increased circulation time in the bloodstream [18]. In order to determine how the length of PEG affects the duration of circulation in the bloodstream of lipoplexes formed by PEG-containing liposomes and RNA, we evaluated the amount of intact siRNA after i.v. injection of siRNA/liposome complexes in the mouse plasma at different time points, using stem-loop PCR. In this experiment, we used 2′-*O*-methylated siMDR, which has no immunostimulating activity and no target in mice, to avoid the stimulation of nonspecific innate immunity. Groups of three CBA/LacSto (*n* = 3) mice were i.v. injected with lipoplexes containing siMDR (0.5 µg/g) and PEGylated or conventional liposomes formed at N/P = 4/1. PEGylated liposomes with the shortest (**P800**) and the longest (**P2000**) PEG-spacers were used.

The results reveal that (Figure 5) the amount of siMDR in the blood plasma administered in complex with **L** liposome was 8.5 pmol/mL 15 min after injection, by 60 min the amount of siMDR sharply decreased to 1.3 pmol/mL. When PEGylated liposomes were used, the concentration of siMDR decreased more slowly: at the first time point studied the concentration of siMDR in the complex with **P800** did not differ from the concentration of siMDR injected in the complex with **L**, but after 60 min its concentration decreased less significantly and exceeded the concentration of siRNA in complex with **L** by 2.8 times and reached 3.7 pmol/mL. The concentration of siMDR injected in the complex with **P2000** exceeded the concentration of siMDR in the complex with other liposomes in just 15 min after the administration, at the 60 min time point it was still five times higher (7 pmol/mL) than that of liposomes **L**. One hundred and twenty min after transfection the concentration of siMDR was below 1 pmol/mL for all liposomes under study. Thus, PEG-containing liposomes significantly increase the circulation time of modified siRNA in the bloodstream in comparison with conventional liposome.

### 2.6. Induction of Cytokines by Immunostimulating RNA Complexed with PEGylated Liposomes in Vivo

Since the study of biodistribution showed that complexes of NA with PEG-containing liposomes stay longer in the bloodstream and accumulate better in the organs, we evaluated their therapeutic potential by the ability to deliver immunostimulating RNA (isRNA) to immune cells and induce cytokine production by these cells in mice. The isRNA under study is one nucleotide longer than canonical small interfering RNAs and has no significant homology with any mRNAs from mice and humans; therefore it does not act via the RNAi mechanism, but induces activation of the innate immune system. We have shown previously [19,20,21] that isRNA exhibits interferon-inducing and antitumor activities in vivo when delivered by liposome L. The activity of lipoplexes, consisting of 10 µg of isRNA and PEGylated liposomes at a concentration corresponding to N/P = 4/1, was estimated by measuring the levels of IFN-α and proinflammatory cytokines IL-6 and TNF-α in the serum of CBA/LacSto mice 6 h after intravenous injection of preparations. The transfection with conventional liposome L was used as a positive control. As a control of specificity of immunostimulating activity of isRNA, siRNA of similar length and structure, but with scrambled sequence (siScr), was used.

The results reveal (Figure 6) that isRNA in complex with both **L** or **P** liposomes efficiently induce IFN-α secretion in mouse blood serum (Figure 6 A). The most effective augmentation of IFN-α level was observed when liposomes with a long PEG spacer, **P1500** and **P2000**, were used, while formulations **L** and **P800** were less effective. Thus, the interferon-inducing activity of isRNA in complexes with **P1500** and **P2000** was ~3.5 times higher than that of complexes with conventional liposome **L**. After the injection of isRNA in complexes with conventional or PEGylated liposomes, TNF-α levels remained unchanged (Figure 6B). The injection of isRNA/**P1500** or **P2000** complexes caused 2–3 fold increases of IL-6 levels relative to the level in animals injected with transfection agent only (to compare under the same conditions, the IFN-α level increased ~70 times). Low level induction of pro-inflammatory cytokines is a favorable factor since these cytokines at high concentrations cause the development of inflammation and related toxicity. It should be noted that control siRNA, siScr, as well as liposomes alone did not induce secretion of IFN-α or cytokines IL-6, TNF-α, indicating the specificity of the isRNA activity.

## 3. Discussion

The physicochemical characteristics and biological properties of PEG-modified cationic liposomes are influenced by the structure of the PEG-containing lipid and its molecular weight [22], as well as its quantitative content in the liposomes [23]. Therefore, in order to select the optimal PEG-containing lipoconjugate for use as a protective shield or a spacer for attachment of the targeting ligands to the earlier developed 2X3-DOPE delivery system [12], it was necessary to vary the length of the PEG chain. 

Apart from PEG length, PEG density is also an important factor affecting the performance of a nanocarrier [24,25]. The quantitative content of PEG-containing lipid in the delivery systems of nucleic acids varies from 0.5 to 30% [26,27,28]. To ensure the most effective protection against interaction with serum proteins, the optimal content of PEG-lipoconjugate is in the range of 2 to 5% [26]. We used 2M% of PEG-containing lipoconjugates in liposomes, which corresponds to the composition of folate-containing liposomes described earlier, with a folate-lipoconjugated PEG (M.W. 800) spacer between folate and the anchor group 2,3-di-*O*-tetradecylglycerol. We showed that the incorporation of such a conjugate into liposomes (2 M%) did not alter the particle size or homogeneity and did not increase the toxicity in a cell culture assay. The same anchor group was used to prepare a series of PEG-lipoconjugates in this study.

The molecular weight of the PEG used to modify delivery systems ranges 350 Da to 40 kDa [29,30] in particular, the most often used lipoconjugates contain PEG residues with 2 kDa M.W. [22,27,31]. The obtained data showed (Table 1) that PEG lipoconjugates with different PEG lengths do not significantly affect the size or homogeneity of the liposomes containing lipoconjugates. A slight decrease in the size of **P** liposomes was observed in comparison with the conventional delivery system **L**; this difference could affect efficiency of the in vivo accumulation in the tissues. It should be noted that lipoplexes formed with siRNA presented more compact particles formed at high N/P ratios in the case of PEG-containing liposomes, than when with **L** liposomes (Table 2). At a low N/P ratio, the changes were non-linear and the particle size decreased in the order **P2000** > **L** > **P1500** > **P800**. With a ratio N/P ≥ 2/1, the ξ potential of lipoplexes becomes positive and also decreases in the order **L** > **P800** > **P1500** > **P2000**.

Investigation of the ability of the obtained liposomes to mediate the delivery of plasmid DNA and siRNA in cell culture has shown that liposomes containing a short PEG-chain lipoconjugate behave in the same way as the liposomes **L**, ensuring effective delivery of therapeutic nucleic acids to target cells and the manifestation of their biological activity both in serum-free medium and in serum-containing medium (Figure 2 and Figure 3). Elongation of the PEG chain leads to a dramatic drop in the efficiency of transfection, and to a corresponding decrease in the biological activity of nucleic acids under the all conditions used [22].

Analysis of the accumulation of siRNA/liposome **P** lipoplexes formed at N/P = 4/1 in mouse organs after i.v. administration showed that the total accumulation in organs has a similar level for all lipoplexes, the highest accumulation of siRNA in the organs of mice was observed after delivery with **P1500** liposomes (Figure 4). This result correlates well with the fact that these liposomes form with siRNA lipoplexes (N/P ratio 4/1) with the smallest size, 93.1 nM.

Biodistribution of siRNA had similar characteristics when liposomes **L**, **P800** and **P1500** were used for delivery, the main difference was observed when using **P2000**: the accumulation in the kidneys and heart increased significantly, while the accumulation in the spleen and liver decreased. 

In order to accumulate in sufficient quantities in organs or in a tumor, lipoplexes must overcome, first of all, a vascular barrier and for this purpose their size should correspond to the properties of the vasculature in the target organ, and the lipoplexes themselves should be present for sufficient time in the bloodstream to ensure effective accumulation. It was demonstrated that liposomes smaller than 100 nm in diameter interacted to a lesser extent with plasma proteins, evaded capture by the RES and had a longer half-life in the blood [32].

PEGylation can significantly improve the pharmacokinetic and pharmacodynamic properties of liposomes and increase their time in the bloodstream, making them invisible to RES [23]. Nevertheless, the properties of PEG-containing liposomes can differ substantially: it was reported, that the half-life time of PEGylated liposomes 250 nm in diameter were about half as long as for the liposomes 100 nm in diameter with similar lipid composition [33]. The data obtained by us showed that the presence of PEG and the length of its chain have a significant effect on the circulation of lipoplexes in the bloodstream, the time of circulation increases significantly with increasing mol. weight of PEG. 

It has been shown for various lipids included in delivery systems for therapeutic agents that they can cause an immunostimulatory effect by alone or intensify the action of other immunostimulators. It was reported that dendritic cells can be targeted precisely and effectively in vivo using intravenously administered RNA-lipoplexes, without the need for functionalization of particles with molecular ligands [34]. Hashimoto et al. demonstrated that immunostimulatory isRNA-containing a PEGylated lipoplex (PEGylated siRNAlipoplex) activates the immune system via TLR7 receptor [35]. Based on these data, we decided to investigate how the length of PEG affects the ability of the liposomes to deliver immunostimulating RNA to immune cells and induce cytokine production by these cells in mice. Earlier we showed that the developed core system **L** effectively delivers isRNA to the immune cells and induces the synthesis of alpha-interferon and the elevation of the level of its production in the case of core system **L** is much higher than when commercially available transfection agent Lipofectamine 2000 was used [19,20,21]. At the same time, the core system L itself does not possess immunostimulating properties. The data obtained by us showed that liposomes with longer PEG-linkers (**P1500** and **P2000**) significantly (up to three times) increase the immunostimulating effect of isRNA without activation of the synthesis of tumor necrosis factor (Figure 5).

Thus, we synthesized PEG-containing lipoconjugates that differed in the molecular weight of PEG (800, 1500 and 2000 kDa). We showed that increasing the length of PEG in the lipoconjugate decreased the effectiveness of the delivery of nucleic acids by the liposomes containing these lipoconjugates in vitro, but improved biodistribution, duration of circulation in the bloodstream and the efficiency of delivery of isRNA in vivo. PEG-containing lipoconjugates obtained in this work can be effectively used to improve the properties of the delivery of isRNAs and also for the design of targeted delivery systems with increased circulating time in the bloodstream.

## 4. Materials and Methods 

### 4.1. General Information

Compounds **1** and **2а** were obtained earlier [5,15]. Poly(ethylene glycol) bis(3-aminopropyl) terminated (1500 Da) and poly(ethylene glycol) bis(amine) (2000 Da) were obtained from Aldrich (St. Louis, MO, USA); other solvents and reagents were purchased from Russian companies (Dia-M, Moscow, Russia; Helicon, Moscow, Russia). CH_2_Cl_2_ and Et_3_N were refluxed with CaH_2_ and distilled prior to the reaction. Pyridine was refluxed with KOH and distilled prior to the reaction. Column chromatography was carried out on silica gel Kieselgel 60 (0.040–0.063 mm or 0.063–0.200 mm, Merck, Darmstadt, Germany). Ion-exchange column chromatography was carried out on Dowex H^+^ (DuPont, Wilmington, DE, USA). ^1^H and ^13^C-NMR spectra were recorded on Avance DPX-300 and Avance DRX-500 pulse Fourier transform spectrometers (Bruker, Karlsruhe, Germany) in CDCl_3_ unless otherwise stated. Chemical shifts were recorded in ppm on the δ scale relative to the CHCl_3_ solvent residual peak (7.26 ppm for ^1^H and 77.0 ppm for ^13^C-NMR spectra). Coupling constants (*J*) are absolute values and recorded in Hz. Mass spectra were run on an Ultraflex time-of-flight mass spectrometer (Bruker, Karlsruhe, Germany) with matrix assisted laser desorption/ ionization (MALDI) and on a Finnigan MAT 900XL-TRAP mass spectrometer (San Jose, CA, USA) with electrospray ionization (ESI). The peak with maximal intensity of PEG were chosen to calculated molecular ion. The spectra exhibited a bell-shaped distribution of 44 Da-spaced lines centered at 2096.615 Da for **2b**, 2138.272 Da for **3b**, 2464.513 Da for **2c**, 2506.662 Da for **3c**. Particle size and zeta potential of liposomes and complexes with nucleic acids were measured using dynamic light scattering (DLS) by a Delsa Nano C particle analyzer (Beckman Coulter, Brea, CA, USA) at 25°C. FACS analysis was performed by a flow cytometer NovoCyte 3000 (ACEA Biosciences, Inc., San Diego, CA, USA).

### 4.2. General Method for the Preparation of Compounds ***2b*** and ***2c***

A solution of diamine (1.3 eq) in anhydrous CH_2_Cl_2_ (4 mL) and anhydrous Et_3_N (4 eq) were added to a solution of *rac*-1-*О*-(4-nitrophenyloxycarbonyl)-2,3-di-*O*-tetradecylglycerol (**1**) (1 eq) in anhydrous CH_2_Cl_2_ (3 mL) and the reaction mixture was stirred at 24 °С for 4 h, and evaporated to dryness in a vacuum. The residue was chromatographed on a cation-exchange column (Dowex H^+^) eluted with MeOH: aq. NH_3_:CHCl_3_ (1:0:0 → 5:1:1) and then on a silica gel column eluted with CHCl_3_:MeOH (20:1 → 10:1). The quality of initial reagents, yield and physico-chemical characteristics of compounds **2b** and **c** are described below.

*O-(3-Ammoniopropyl)-O′-[3-(rac-2,3-di(tetradecyloxy)prop-1-yloxycarbonyl) aminopropyl] poly(ethylene glycol_1500_) hydrochloride* (**2b**): *rac*-1-*О*-(4-nitrophenyloxycarbonyl)-2,3-di-*O*-tetradecylglycerol (182 mg, 0.280 mmol), poly(ethylene glycol) bis(3-aminopropyl) terminated (1500 Da) (650 mg, 0.433 mmol), Et_3_N (150 µL). Compound **2b** was obtained at 20% yield (0.115 g) as a white crystallizing oil. ^1^H-NMR NMR (300 MHz, CDCl_3_-CD_3_OD, 6:1): 0.81 (t, *J* = 6.7 Hz, 6H, 2 (CH_2_)_11_Me,); 1.11–1.29 (m, 44H, 2 (CH_2_)_11_Me); 1.44–1.54 (m, 4H, 2 OCH_2_CH_2_); 1.66–1.75 (m, 2H, NH_2_CH_2_CH_2_); 1.84–1.93 (m, 2H, NHCH_2_CH_2_); 3.14–3.20 (m, 2H, CH_2_NH_2_); 3.26–3.32 (m, 2H, NHCH_2_); 3.34–3.73 (m, 143H, CH_2_(OCH_2_CH_2_)_33_OCH_2,_ 2 OCH_2_CH_2_, OCH_2_CHO); 3.96–4.11 (m, 2H, CH_2_OC(O)). ^13^С-NMR (125 MHz, CDCl_3_): 8.35, 13.75, 22.39, 25.76, 25.81, 26.06, 29.07, 29.21, 29.36, 29.40, 29.72, 31.64, 38.19, 39.76, 42.62, 46.03, 63.82, 68.79, 68.85, 69.60, 69.73, 69.77, 69.84, 69.92, 69.96, 70.03, 70.19, 70.37, 71.55, 76.70, 156.63. MS (MALDI) *m*/*z*: [M − HCl + H]^+^ calcd for C_104_H_211_N_2_O_38_, 2096.464; found: 2096.615.

*O-(2-Ammonioethyl)-O′-[2-(rac-2,3-di(tetradecyloxy)prop-1-yloxycarbonyl)aminoethyl] poly(ethylene glycol_2000_) hydrochloride* (**2с**): *rac*-1-*О*-(4-Nitrophenyloxycarbonyl)-2,3-di-*O*-tetradecylglycerol (47 mg, 0.072 mmol), poly(ethylene glycol) bis(amine) (2000 Da) (208 mg, 0.104 mmol), Et_3_N (50 µL). The compound **2c** was obtained in 38% yield (0.068 g) as white crystallizing oil. ^1^H-NMR (300 MHz, CDCl_3_): 0.81 (t, *J* = 6.7 Hz, 6H, 2 (CH_2_)_11_Me,); 1.10–1.30 (m, 44H, 2 (CH_2_)_11_Me); 1.43–1.53 (m, 4H, 2 OCH_2_CH_2_); 3.07–3.15 (m, 2H, CH_2_NH_2_); 3.24–3.33 (m, 2H, NHCH_2_); 3.33–3.75 (m, 179H, CH_2_(OCH_2_CH_2_)_42_OCH_2,_ 2 OCH_2_CH_2_, OCH_2_CHO); 4.01 (dd, *J* = 5.4, *J* = 11.6, 1H) and 4.08–4.15 (m, 1H, CH_2_OC(O)); 5.15–5.22 (m, 1H, CONH). MS (MALDI) *m*/*z*: [M – HCl + H]^+^ calcd for C_120_H_243_N_2_O_47,_ 2464.669; found: 2464.513.

### 4.3. The General Method for the Preparation of Compounds ***3a***–***c***

Anhydrous pyridine (5 eq) and acetic anhydride (5 eq) were added to a solution of compounds **2а**–**с** (1 eq) in anhydrous CH_2_Cl_2_ (2 mL) and the reaction mixture was stirred at 24 °С for 6 hours. The solvent was evaporated to dryness in a vacuum. The product was isolated by ion-exchange column chromatography (MeOH → MeOH – aq. NH_3_, 5:1). The quality of initial reagents, yield and physico-chemical characteristics of compounds **3** are described below.

*O-(2-(Acet)amidoethyl)-O′-[2-(rac-2,3-di(tetradecyloxy)prop-1-yloxycarbonyl)aminoethyl] octadecaethylene glycol* (**3а**): compound **2a** (38.7 mg, 0.0275 mmol), acetic anhydride (36 µL), pyridine (36 µL). The compound **3a** was obtained in 88% yield (0.035 g) as white crystallizing oil. ^1^H-NMR (500 MHz, CDCl_3_): 0.81 (t, *J* = 6.7 Hz, 6H, 2 (CH_2_)_11_Me); 1.10–1.29 (m, 44H, 2 (CH_2_)_11_Me); 1.44–1.57 (m, 4H, 2 OCH_2_CH_2_); 1.91(s, 3H, CH_3_COO); 3.25–3.68 (m, 86H, 2 NHCH_2_, CH_2_(OCH_2_CH_2_)_18_OCH_2,_ 2 OCH_2_CH_2_, OCH_2_CHO); 3.68–3.77 (m, 1H, OCH_2_CHO); 4.02 (dd, *J* = 5.4 Hz, *J* = 11.4 Hz, 1H) and 4.07–4.15 (m, 1H, CH_2_OC(O)); 5.21–5.30 (m, 1H, CONH); 6.46–6.56 (m, 1H, CONH). ^13^С-NMR (125 MHz, CDCl_3_): 14.04, 22.61, 25.99, 26.04, 29.28, 29.43, 29.58, 29.61, 29.98, 31.85, 39.26, 40.82, 69.80, 70.02, 70.14, 70.27, 70.51, 71.71, 170.28. MS (MALDI) *m*/*z*: [M + H]^+^ calcd for C_74_H_149_N_2_O_24_, 1450.050; found: 1449.042.

*O-(3-(Acetamidopropyl)-O′-[3-(rac-2,3-di(tetradecyloxy)prop-1-yloxycarbonyl)aminopropyl] poly(ethylene glycol_1500_)* (**3b**): compound **2b** (50.9 mg, 0.0243 mmol), acetic anhydride (24 µL), pyridine (20 µL). The compound **3b** was obtained in 77% yield (0.040 g) as white crystallizing oil. ^1^H-NMR (300 MHz, CDCl_3_): 0.81 (t, *J* = 6.7 Hz, 6H, 2 (CH_2_)_11_Me,); 1.13–1.29 (m, 44H, 2 (CH_2_)_11_Me); 1.43–1.52 (m, 4H, 2 OCH_2_CH_2_); 1.66–1.75 (m, 4H, 2 NHCH_2_CH_2_); 1.88 (s, 3H, CH_3_COO); 3.17–3.25 (m, 2H, NHCH_2_); 3.26–3.32 (m, 2H, NHCH_2_); 3.33–3.63 (m, 142H, CH_2_(OCH_2_CH_2_)_33_OCH_2,_ 2 OCH_2_CH_2_, OCH_2_CHO); 3.69–3.73 (m, 1H, OCH_2_CHO); 4.01 (dd, *J* = 5.2, *J* = 11.5, 1H) and 4.06–4.14 (m, 1H, CH_2_OC(O)); 5.07–5.14 (m, 1H, CONH); 6.26–6.35 (m, 1H, CONH). MS (MALDI) *m*/*z*: [M + H]^+^ calcd for C_106_H_213_N_2_O_39,_ 2138.475; found: 2138.272.

*O-(2-(Acetamidoethyl)-O′-[2-(rac-2,3-di(tetradecyloxy)prop-1-yloxycarbonyl)aminoethyl] poly(ethylene glycol_2000_)* (**3с**): compound **2c** (23.3 mg, 0.0094 mmol), acetic anhydride (50 µL), pyridine (50 µL). The compound **3c** was obtained in 80% yield (0.019 g) as white crystallizing oil. ^1^H-NMR (300 MHz, CDCl_3_-CD_3_OD, 6:1): 0.81 (t, *J* = 6.7 Hz, 6H, 2(CH_2_)_11_Me,); 1.12–1.32 (m, 44H, 2(CH_2_)_11_Me); 1.44–1.55 (m, 4H, 2 OCH_2_CH_2_); 1.91 (s, 3H, CH_3_COO); 3.21–3.68 (m, 182H, 2 NHCH_2_, CH_2_(OCH_2_CH_2_)_42_OCH_2,_ 2 OCH_2_CH_2_, OCH_2_CHO); 3.68–3.74 (m, 1H, OCH_2_CHO); 4.01 (dd, *J* = 5.4, *J* = 11.6, 1H) and 4.06–4.13 (m, 1H, CH_2_OC(O)). ^13^С-NMR (125 MHz, CDCl_3_): 13.84, 22.36, 22.52, 25.94, 29.20, 29.35, 29.50, 29.54, 31.78, 39.20, 69.53, 69.93, 70.33, 71.71, 109.20, 160.10. MS (MALDI) *m*/*z*: [M + H]^+^ calcd for C_122_H_245_N_2_O_48,_ 2506.679; found: 2506.662. 

### 4.4. Liposome Preparation

All liposomal formulations were prepared by hydrating thin lipid film method [12]. Briefly, a solution of the polycationic lipid 1,26-bis(cholest-5-en-3β-yloxycarbonylamino)-7,11,16,20-tetraazahexacosan tetrahydrochloride (2X3) [13] in a mixture of CHCl_3:_CH_3_OH (1:1 mol.) was added to a solution of dioleoylphosphatidylethanolamine (DOPE, Avanti Polar Lipids, Alabaster, AL, USA) in CHCl_3_ at a molar ratio of 1:1, and gently stirred. A solution of lipoconjugate **3a**–**c** (2 M%) in CHCl_3_:CH_3_OH (1:1) was added to the 2X3-DOPE mixture, and organic solvents were removed in vacuo. The obtained lipid film was dried for 4 h at 0.1 Torr to remove residual organic solvents and was hydrated in deionized water (MilliQ, Burlington, MA, USA) at 4°C overnight. The resulting liposomal dispersion was sonicated for 15 min at 70–75 °C in a bath-type sonicator (Bandelin Sonorex Digitec DT 52H, Berlin, Germany) flushed with argon and stored at 4 °C. In the resulting dispersion, the cationic lipid 2X3 concentration was 1 mM.

### 4.5. Liposome Sizes and Zeta Potentials 

Particle size and zeta potential was measured using a Delsa Nano dynamic light scattering instrument with zeta-sizing capacity (Beckman Coulter, Brea, CA, USA). For lipoplex characterization, 50 μL of nucleic acid solution prepared in MilliQ water was mixed with 50 μL of liposome solution taken at the appropriate N/P ratio. Then, after 20 min of incubation at room temperature, the analysis of size was performed using a 100 μL microcuvette. For zeta-potential determination, 900 μL of MilliQ water was added to the sample and surface potential was recorded in a 1 mL cuvette. 

### 4.6. Plasmid and RNA

pEGFP-C2 plasmid (Clontech, Heidelberg, Germany) was used for the plasmid DNA (pDNA) transfection experiments. The strands of small interfering RNA (siRNA) or immunostimulating RNA (isRNA) were synthesized on an automatic ASM-800 synthesizer (Biosset, Novosibirsk, Russia) using solid-phase phosphoramidite synthesis protocols [36] optimized for the instrument, with a 10 min coupling step for 2′-*O*-TBDMS-protected phosphoramidites and a 6 min coupling step for 2′-*O*-methylated phosphoramidites. A C6 CPG 3′-PT-amino-modifier (Glen Research, Sterling, VA, USA) was used for synthesis of the 3′-aminohexyl-containing strand. The following siRNAs and isRNA were used in the present study (2′-*O*-methyl-modified C and U nucleotides are designated as Cm and Um): siEGFP targeted to *EGFP* mRNA (sense strand 5′-GAACGGCAUCAAGGUGAACTT-3′; antisense strand 5′-GUUCACCUUGAUGCCGUUCTT-3′ [37]; siScr with no significant homology to any known mouse, rat or human mRNA sequence (sense strand 5′-CAAGUCUCGUAUGUAGUGGUU-3′; antisense strand 5′-CCACUACAUACGAGACUUGUU-3′); siMDR targeted to *MDR1* mRNA (sense strand 5′- GGCUUmGACmAAGUUmGUmAUmAUmGG-3′; antisense strand 5′-AUmAUmACmAACUU mGUCmAAGCCmAA-3′) [38,39] and isRNA (strand 1: 5′-AAAUCUGAAAGCCUGACACUUA-3′ and strand 2: 5′-GUGUCAGGCUUUCAGAUUUUUU-3′) [19]. Cyanine5.5 (Cy5.5) was attached to the 3′-end of the antisense strand of siMDR equipped with a 3′-aminohexyl linker according to the manufacturer’s protocol, using Cy5.5 *N*-hydroxysuccinimide esters (Biotech Industry Ltd., Moscow, Russia) in 0.1 M Tris buffer (pH 8.4). Isolation of the oligoribonucleotides and the conjugate was accomplished by electrophoresis in a 12% denaturing polyacrylamide gel (dPAAG). The purified oligoribonucleotides were characterized by electrophoretic mobility in 12% dPAAG and by MALDI-TOF-MS. siRNAs and isRNA (50 μM) were annealed in a buffer containing 30 mM HEPES-KOH (pH 7.4), 100 mM sodium acetate and 2 mM magnesium acetate, by heating at 90 °C for 5 min, followed by cooling to room temperature, and stored at −20 °C until required.

### 4.7. Preparation of Complexes of Cationic Liposomes and Nucleic Acids

Prior to use, the cationic liposome and nucleic acid complexes were formed in serum-free Opti-MEM (Invitrogen, Waltham, MA, USA) by mixing of equal volumes of liposomes and respective nucleic acid solutions in Opti-MEM taken at appropriate concentrations corresponding to a final concentration for in vitro experiments of 2 μg/mL for pEGFP-C2, 50 nM for siEGFP in the well or, for in vivo experiments, of 1 µg/g for Cy-5.5-siMDR, 0.5 µg/g for isRNA; the resulting mixtures were incubated for 20 min at room temperature. NA/liposome complexes were formed at N/P ratios of 4/1, 6/1, and 8/1; N/P ratios are indicated in corresponding figure legends.

### 4.8. Cell Lines and Culture Conditions

Genetically modified BHK IR780 cells, constantly expressing EGFP, derived from BHK (baby hamster kidney) cells were kindly provided by Prof. V. Prasolov (Engelhardt Institute of Molecular Biology, RAS, Moscow, Russia). HEK 293 (human embryonic kidney), BHK and BHK IR780 cells were grown in DMEM (Sigma, St. Louis, MS, USA) supplemented with 10% fetal bovine serum (FBS, GE Healthcare, Chicago, IL, USA), 100 μg/mL penicillin, 100 μg/mL streptomycin and 0.25 μg/mL amphotericin at 37°C in a humidified atmosphere containing 5% CO_2_/95% air. 

### 4.9. Cell Viability Assay

The relative amount of living cells after incubation with liposomes was determined by the MTT test [16]. HEK 293 cells were plated in 96-well culture plates (10^4^ cells/well) and incubated with cationic liposomes (final concentration in the well 1–80 μM) for 4 h under serum-free conditions, then serum was added to each well and cells were incubated for a further 20 h. After 24 h of incubation, a solution of MTT (3-(4,5-dimethylthiazolyl)-2,5-diphenyltetrasolium bromide) (Sigma, St. Louis, MS, USA) was added to a final concentration of 0.5 mg/mL, and cells were incubated for an additional 3 h. The culture medium was then removed, formosan crystals were solubilized in DMSO, and the differences in absorbance at 570 and 620 nm were measured spectrophotometrically using a Multiscan RC (Labsystems, Vantaa, Finland). Results are expressed as mean values of measurements for three wells ± SD.

### 4.10. Transfection of Cells

HEK 293 (1.2 × 10^5^ cells/well) and BHK IR780 (0.15 × 10^5^ cells/well) cells were seeded in 24-well plates one day before transfection. On the day of the experiment, the culture medium of cells was replaced with 200 μL of fresh serum-free or medium supplemented with 10% FBS (see figure legends). The cationic liposome/nucleic acid complexes were pre-formed in serum-free Opti-MEM medium as described above. The lipoplexes were formed at an NA concentration corresponding to a final concentration in the well of 2 μg/mL for pEGFP-C2 and 50 nM for siEGFP. Following pre-incubation at room temperature, the lipoplexes were added to the cells, which were incubated under standard conditions for 4 h. Then the culture medium was replaced with 500 µL of DMEM supplemented with 10% FBS. The cells were allowed to grow under standard conditions for an additional 44 h in the case of pEGFP-C2 delivery or 68 h in the case of siRNA delivery. The level of pEGFP-C2 delivery was evaluated 48 h post-transfection by measuring EGFP expression. The efficiency of siEGFP delivery was determined by inhibition of EGFP expression in BHK IR780 cells measured 72 h after transfection. EGFP expression in the cells was analyzed using a Cytomics FC 500 (Beckman Coulter, Brea, CA, USA) flow cytometer (excitation wave length 488 nm, emission 530 ± 30 nm). 10,000 cells from each sample were analyzed.

### 4.11. Mice

All animal procedures were carried out in strict accordance with the recommendations for proper use and care of laboratory animals (ECC Directive 86/609/EEC). The protocol was approved by the Committee on the Ethics of Animal Experiments of the Administration of the Siberian Branch of the Russian Academy of Sciences. The experiments were conducted at the Center for Genetic Resources of Laboratory Animals at the Institute of Cytology and Genetics, Siberian Branch, Russian Academy of Sciences (RFMEFI61914X0005 and RFMEFI62114Х0010). 8- to 10-week-old female SCID (SHO-PrkdcscidHrhr) mice with an average weight of 20–22 g from the Center for Genetic Resources of Laboratory Animals at the Institute Cytology and Genetics SB RAS were used. Also we used 8–10-week-old male CBA/LacSto (CBA) mice with an average weight of 23–27 g from the vivarium of the Institute of Cytology and Genetics SB RAS. Mice were housed in groups of 8–10 individuals in plastic cages, and had free access to food and water; daylight conditions were normal.

### 4.12. In Vivo Biodistribution Studies 

In vivo real-time fluorescence imaging was used to evaluate the distribution of Cy5.5-labeled siRNA in SCID mice. The In-vivo MS FX PRO Imaging System (Carestream, Rochester, NY, USA) was used to obtain X-rays and, concurrently, near-infrared fluorescence (NIRF) images (Cy5.5: excitation 675 nm, emission 694 nm). Mice were injected intravenously (i.v.) with 1 µg/g Cy5.5-labeled siMDR complexed with cationic liposomes **L** (2X3-DOPE), **P800**, **P1500** or **P2000** (containing lipoconjugates **3a**–**c**, accordingly) at N/P ratio 4/1 in 200 μL Opti-MEM medium. The dose of siRNA was 1 µg/g of mouse body weight. Animals were anesthetized with avertin (150 mg/kg, intraperitoneally) and placed on a heating tray (37 °C). The fluorescence (10 s exposure) and X-ray (15 s exposure) scans were performed at different time points (20 min and 24 h) post-injection. At the end of the experiment, the mice were euthanized and brains, lungs, hearts, livers, spleens and kidneys were collected. Each organ was rinsed with PBS and the fluorescence intensity detected. Fluorescence intensities of organs were measured by creating an automatic ROI with a threshold of 30% of each sample′s maximum intensity, and measuring that area′s mean by multiplying each luminance level by the number of pixels at that level, and then dividing by the total number of gray levels. Fluorescence intensities were normalized to the peak angle of detection and the area of the organ, and figures were then created in Origin 6.1 (v 6.1052, OriginLab, Northampton, MA, USA). Images were batch exported as 16-bit TIFFs, and overlays were completed in Adobe Photoshop CS3 (v 10.0, Adobe Systems, San Jose, CA, USA). Data was obtained from three independent sets of experiments with identical experimental set-ups.

### 4.13. Quantitative Stem-Loop Real-Time PCR Analysis

CBA/LacSto mice were i.v. injected with 0.5 µg/g isRNA/liposome complexes (N/P = 4/1) and blood plasma samples were collected 15, 60, 120 and 180 min after the last injection. The RNA from plasma samples was isolated according to Landesman et al. [40]. Briefly, the plasma samples were diluted 1:10 in 0.25% Triton X-100 (PanReac AppliChem, Barcelona, Spain), followed by heating at 95 °С for 10 min. Then, the samples were cooled on ice for 10 min, centrifuged at 13,000 rpm for 20 min at 4 °С and supernatants were collected and kept on ice. 

siRNA specific stem-loop qRT-PCR assays were designed according to the instructions of Czimmerer et al. [41] using UPL-probe based stem-loop quantitative PCR assay design software (Astrid Research, Debrecen, Hungary) (http://genomics.dote.hu:8080/mirnadesigntool).

The reverse transcription reaction was performed in a final volume of 40 µL containing 2 µL of Triton X-100 preheated plasma supernatant, 4 µL 10 µM siMDR specific stem loop-RT primer (5′-GTTGGCTCTGGTGCAGGGTCCGAGGTATTCGCACCAGAGCCAACTTGGCT-3′) and 34 µL of Master Mix prepared from a M-MuLV–RH Reverse Transcription Kit (Biolabmix, Novosibirsk, Russia). The reaction mixture was incubated at 42 °C for 1 h. The cDNA was amplified in a total volume of 20 µL containing 5 µL of cDNA template, siMDR specific forward (5’-GTTGGGGATATACAACTTGTCA-3′) and reverse (5′-GTGCAGGGTCCGAGGT-3’) primers (1 µM) and BioMaster HS-qPCR SYBR Blue master mix with SYBR Green I fluorescent dye (Biolabmix, Novosibirsk, Russia) using an IQ5 Cycler (Bio-Rad, Hercules, CA, USA). All measurements were done in quadruplicate.

### 4.14. Analysis of the Level of Cytokines in Mouse Blood Serum

CBA/LacSto mice were i.v. injected with conventional liposome **L** or liposomes P800, P1500 or P2000 pre-complexed with isRNA (0.5 µg/g, N/P = 4/1), and blood was collected 6 h after injection via head clipping. The serum was prepared from the whole blood by coagulation for 30 min at 37 °C and subsequent centrifugation. The concentrations of IFN-α, TNF-α and IL-6 in the serum were determined by sandwich ELISA kits (BD Biosciences, San Jose, CA, USA) in accordance with the manufacturer’s instructions. The levels of cytokines was measured in duplicate in individual serum samples from three mice per group.

### 4.15. Statistical Analyses

The variables were expressed as mean ± standard deviation (SD). Mean values were considered significantly different when *p* < 0.05, using Student’s t test. 

## 5. Conclusions

We demonstrate that increasing the length of the PEG chain reduces the transfection activity of liposomes in vitro, but improves the biodistribution, increases the circulation time in the bloodstream and enhances the interferon-inducing activity of immunostimulating RNA in vivo.

## Figures and Tables

**Figure 1 molecules-23-03101-f001:**
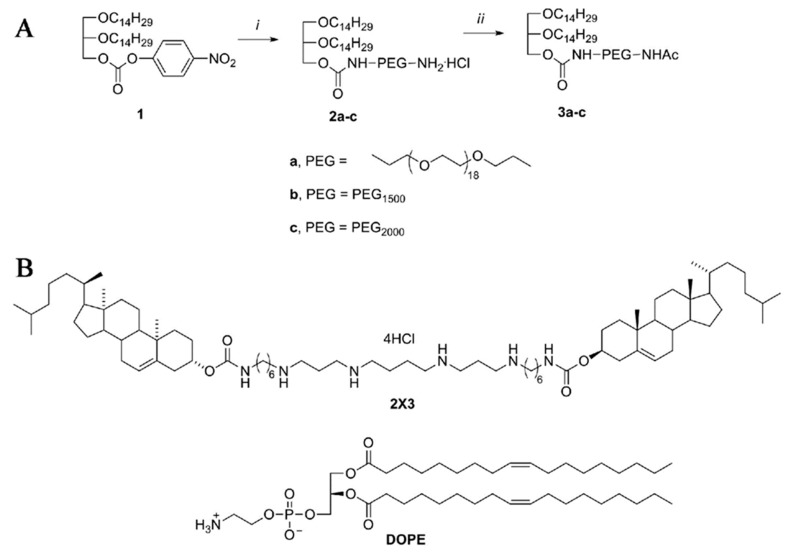
Scheme of synthesis of lipoconjugates **3a**–**c** (**a**–**c** show the structure of spacer used) (**A**) and structures of cholesterol-based polycationic amphiphile 2X3 and lipid helper DOPE used for liposomes preparation (**B**). Reagents and conditions: (*i*) 4-NO_2_C_6_H_4_OCOCl and Et_3_N, 24 °С, 11 h, then H_2_N-PEG-NH_2_, 24 °С, 1 h, 57–73%; (*ii*) Ac_2_O, Py, 24 °C, 6 h, 77–88%.

**Figure 2 molecules-23-03101-f002:**
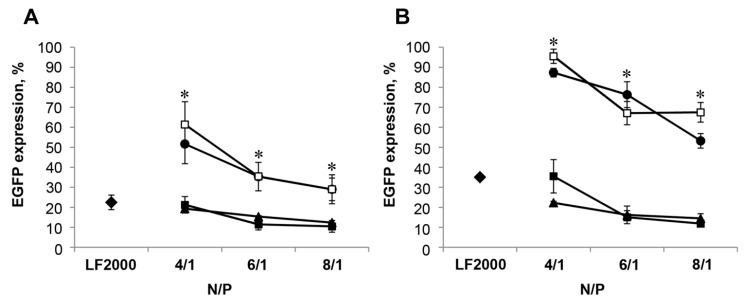
Silencing of EGFP gene expression in BHK IR780 cells by siEGFP delivered by cationic liposomes in serum-free conditions (**A**) or in the presence of 10% FBS (**B**). The cells were incubated with the lipoplexes formed by cationic liposomes (■ for L, ▲ for **P800**, ● for **P1500** and □ for **P2000**) and siEGFP (50 nM) at various N/P ratios. The levels of EGFP expression in the cells were evaluated using flow cytometry after 72 h of incubation with the lipoplexes. Levels of EGFP expression in the control cells (without any treatment) was set at 100%. The differences between mean values for L and other preparations at the same N/P were considered statistically significant: * *p* < 0.01 (Mann-Whitney U-test). siEGFP delivery by Lipofectamine 2000 (♦ for **LF2000**) (under optimal conditions) was used as a positive control.

**Figure 3 molecules-23-03101-f003:**
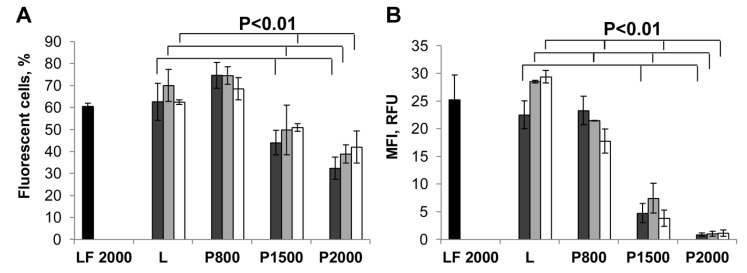
Liposomes-mediated delivery of plasmid DNA in HEK 293 cells measured by EGFP expression. The percentage of EGFP-positive cells (**A**) and mean fluorescence intensity (MFI) of the cell population (**B**) were evaluated by flow cytometry 48 h after transfection. Cells were incubated with the lipoplexes formed by liposomes and pEGFP-C2 (2 μg/mL) at N/P ratios of 4/1 (dark grey bars), 6/1 (light grey bars) or 8/1 (white bars) in the presence of 10% FBS. **LF2000**- Lipofectamine 2000 (black bars) was used under optimal conditions. The differences between mean values for L and other preparations at the same N/P with *p* < 0.01 were considered statistically significant according to Mann-Whitney U-test.

**Figure 4 molecules-23-03101-f004:**
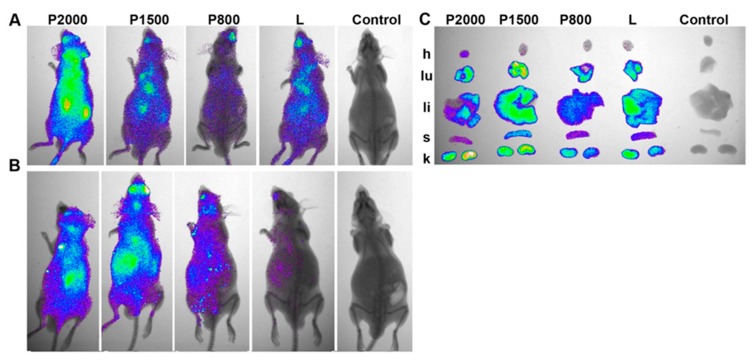
Biodistribution of Cy5.5-labeled siMDR complexed with liposomes **P800**, **P1500**, **P2000** or **L** in SCID mice. (**A**,**B**) Lifetime fluorescence imaging of SCID mice (dorsal orientation view) 20 min (exposure time 20 s) (**A**) or 24 h (exposure time 60 s) (**B**) after i.v. injection. (**C**) Images of dissected organs of SCID mice at 24 h time point (h: heart, lu: lungs, li: liver, s: spleen, k: kidney). **L**/siRNA and **P**/siRNA lipoplexes were formed at N/P = 4/1; control: non-injected mice.

**Figure 5 molecules-23-03101-f005:**
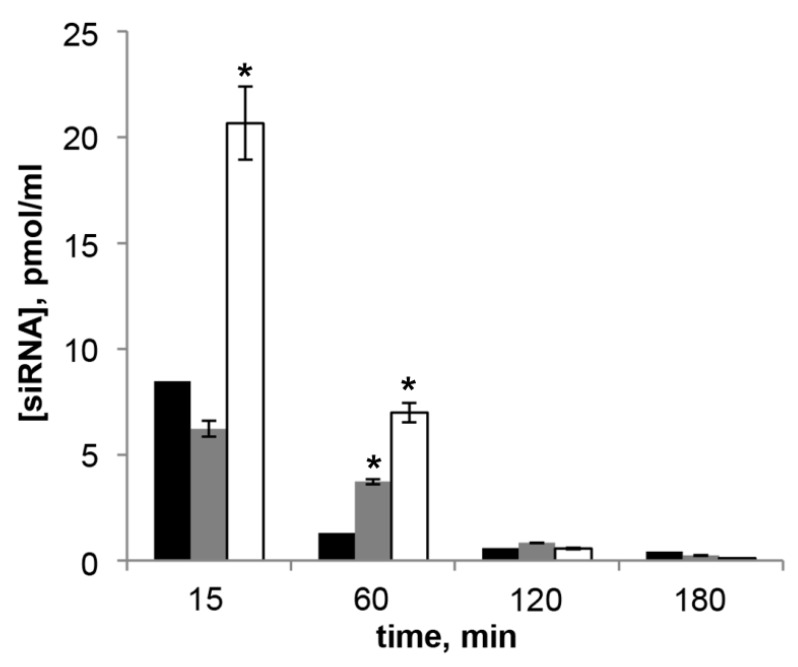
Concentration of siMDR measured by stem-loop PCR in blood plasma of CBA/LacSto mice 15–180 min after i.v. injection of **L**/siMDR (black bars), **P800**/siMDR (grey bars) or **P2000**/siMDR (white bars) lipoplexes formed at N/P = 4/1. The data represent means ± standard deviation (SD) (*n* = 3). The differences between mean values for **L** and other preparations at the same N/P were considered statistically significant * *p* < 0.01 (Mann-Whitney U-test).

**Figure 6 molecules-23-03101-f006:**
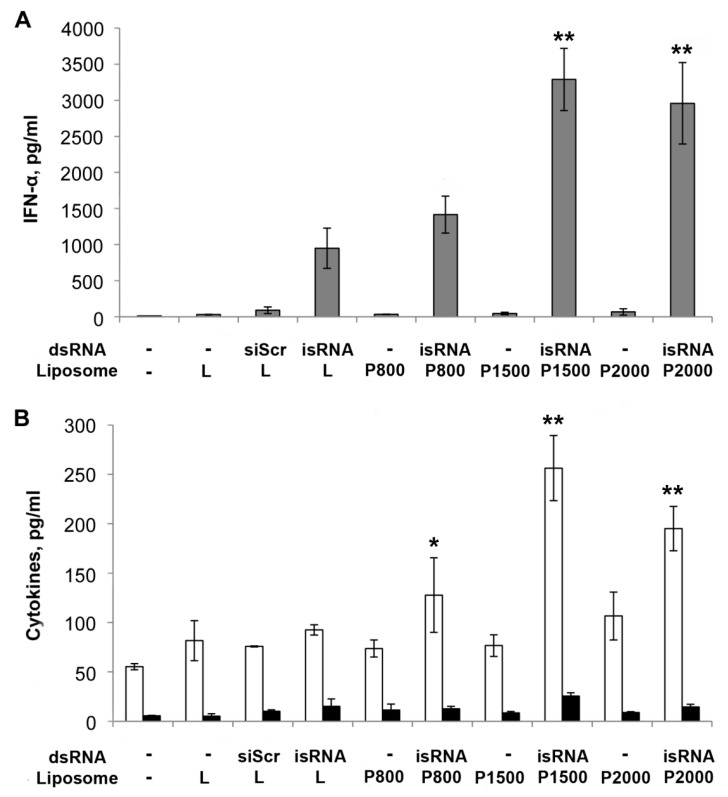
Stimulation of the innate immune response by isRNA in complexes with PEGylated **P** and conventional **L** liposomes in vivo. СBA/LacSto mice were injected i.v. with isRNA (10 μg per mouse) in complex with liposomes (at the ratio N/P = 4/1). siRNA with scrambled sequence (siScr) was used as a control. (**A**) Serum IFN-α and (**B**) IL-6 (white columns) and TNF-α (black columns) levels were measured by ELISA 6 h after injection. The data represent means ± standard deviation (SD) calculated from measurements from at least three mice. The differences between mean values for isRNA/**L** and other preparations were considered statistically significant: * *p* < 0.05, ** *p* < 0.01 (Mann-Whitney U-test).

**Table 1 molecules-23-03101-t001:** Hydrodynamic diameters, polydispersity index and cytotoxicity of PEG-containing cationic liposomes.

Cationic Liposome ^1^	PEG Lipoconjugate	Diameter, Mean, nm	PI ^2^	Living Cells ^3^, %
**L**	-	95.7 ± 19.5	0.260 ± 0.055	72.2 ± 4.5
**P800**	**3a**	67.3 ± 15.3	0.207 ± 0.015	64.1 ± 5.6
**P1500**	**3b**	76.5 ± 13.5	0.226 ± 0.014	60.1 ± 1.1
**P2000**	**3c**	50.6 ± 12.7	0.245 ± 0.006	79.7 ± 13.2

^1^ Conventional liposomes 2X3:DOPE prepared at 1:1 molar ratio were designated as **L**. PEG-containing liposomes were designated as P. ^2^ PI: polydispersity index. ^3^ The amount of live HEK293 cells in the population after incubation in the presence of 80 μM liposomes for 24 h, MTT (3-(4,5-dimethylthiazol-2-yl)-2,5-diphenyltetrazolium bromide) assay data.

**Table 2 molecules-23-03101-t002:** Hydrodynamic diameters, polydispersity index and ξ-potentials of lipoplexes formed by siRNA (siScr) and PEG-containing (**P**) or conventional (**L**) cationic liposomes.

Cationic Liposomes	N/P	Diameter, nm	Polydispersity Index (PI)	ξ-Potential, mV
**L**	1/1	2298.7 ± 320.4	0.459 ± 0.116	14.9 ± 1.8
	2/1	276.1 ± 5.3	0.237 ± 0.038	46.4 ± 1.8
	4/1	206.0 ± 10.2	0.267 ± 0.042	45.3 ± 1.7
**P800**	1/1	246.5 ± 8.9	0.528 ± 0.049	−23.1 ± 1.3
	2/1	175.2 ± 22.6	0.305 ± 0.027	44.6 ± 1.2
	4/1	178.4 ± 3.4	0.334 ± 0.043	40.2 ± 0.6
**P1500**	1/1	836.1 ± 65.8	0.337 ± 0.021	30.1 ± 2.6
	2/1	173.3 ± 6.8	0.353 ± 0.061	35.2 ± 3.6
	4/1	93.1 ± 4.1	0.235 ± 0.030	39.6 ± 1.3
**P2000**	1/1	4038.3 ± 736.1	0.625 ± 0.264	−1.3 ± 0.2
	2/1	180.2 ± 8.4	0.303 ± 0.055	32.8 ± 1.4
	4/1	143.0 ± 7.1	0.328 ± 0.087	37.5 ± 3.2

**Table 3 molecules-23-03101-t003:** Accumulation of Cy5.5-labeled siMDR complexed with liposome **P800**, **P1500**, **P2000** or **L** formed at N/P of 4/1 in the organs of healthy SCID mice, 24 h after i.v. injection.

Organ	Total Fluorescence of the Organ, RFU ^a^				Total Florescence in the Organ Relative to the Total Fluorescence in All Organs, %			
	P2000	P1500	P800	L	P2000	P1500	P800	L
Heart	44.9 ± 1.9	14.1 ± 0.7	6.9 ± 0.4	9.7 ± 0.4	10.6 ± 0.4	2.6 ± 0.1	2.2 ± 0.1	2.5 ± 0.1
Lungs	112 ± 5.7	161.3 ± 9.2	85.3 ± 5.2	107.4 ± 5.8	26.5 ± 1.3	29.7 ± 1.7	27.8 ± 1.7	27.4 ± 1.5
Liver	65.6 ± 3.2	121 ± 7.1	72.6 ± 3.6	108.3 ± 5.7	15.5 ± 0.8	22.3 ± 1.3	23.7 ± 1.2	27.6 ± 1.5
Spleen	31.4 ± 1.4	90.7 ± 3.9	45.2 ± 3.3	53 ± 3.1	7.4 ± 0.3	16.7 ± 0.7	14.7 ± 1.1	13.5 ± 0.8
Kidneys	169.3 ± 9.5	155.8 ± 8.9	96.8 ± 5.8	114.4 ± 4.2	40 ± 2.2	28.7 ± 1.6	31.5 ± 1.9	29.1 ± 1.1
Total	423.2 ± 21.7	542.9 ± 29.8	306.8 ± 18.3	392.8 ± 19.2	100	100	100	100

Mean ± SD (*n* = 3). ^a^ RFU: relative fluorescence units.

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
