# Peer review of "Novel PEGylated Liposomes Enhance Immunostimulating Activity of isRNA"

_molecules, 2018, doi:10.3390/molecules23123101_

Round 1
Reviewer 1 Report
Add statistical significance in the figures 2, 3, 5 and 6.
Tissue specific distribution of the polymers is apparent (figure 4) therefore using another cell line apart from HEK-293 to carry out the experiments shown in figure 3 would be informative.
Page 7, line 253: Indicate the number of animals per group.
Page 8, line 273: In the legend of the figure 5 it is mentioned that grey bars represent P1500/siMDR whereas in the text (page 7, lines 254 and 255) it is mentioned that “PEGylated liposomes with the shortest (P800) and the longest (P2000) PEG-spacers were used.”
Page 8, lines 296 and 297: “After the injection of isRNA in complexes with conventional or PEGylated liposomes, TNF-α levels remained unchanged (Fig. 6B).” The size of the IL-6 bars masks the effect of isRNA-P1500 on TNF-α levels. Please consider presenting the TNF-α in a separate diagram from the IL-6. Please indicate statistical significance.
Minor points
Consider using dispersity index (Đ) instead of polydispersity index (PI)
Page 4, line 120: number instead of amount of living cells
Page 10, line 319: Add reference where indicated.
Page 1, line 43: “Cationic lipoplexes, which are a highly effective” delete the word a.
Page 10, line 347: “…under the all conditions used.” delete the word the.
Page 11, line 371: “…cause an immunostimulatory effect by alone” delete the word by.
Page 13, line 482: lipoconjugate instead of lipoconjugat.
Author Response
Dear Reviewer #1,
Thank you for careful study of our manuscript and for your very useful remarks and comments. We revised the manuscript and let us respond to your questions and comments.
REW1
Add statistical significance in the figures 2, 3, 5 and 6.
Corrected, the evaluation of statistical significance was added in the figures 2,3,5 and 6.
Tissue specific distribution of the polymers is apparent (figure 4) therefore using another cell line apart from HEK-293 to carry out the experiments shown in figure 3 would be informative.
In vivo distribution of the lipoplexes formed by Cy5.5-labeled siMDR complexed with cationic liposomes P800, P1500, P2000 or L similar to a number of different other nanocomplexes depends rather on the level of organ or tissue vascularization than on type of cells forming this tissue (organ). Our previous results (see refs 12-14 and 19-21, this manuscript) clearly show that core liposomal formulation L provide efficient delivery of various therapeutic nucleic acids (plasmid DNA, antisense oligonucleotides, siRNAs and immunostimulatory RNA) in cells in vitro exhibiting similar dependencies of transfection efficiency on N/P ratios. It worth mentioning that differences in transfection efficiency (accumulation level) observed in vitro for different cell types (tumor cells, dendritic cells, PBMC cells etc) did not correlate directly with observed biological effects of the same formulations in vivo.
Page 7, line 253: Indicate the number of animals per group.
Corrected.
Page 8, line 273: In the legend of the figure 5 it is mentioned that grey bars represent P1500/siMDR whereas in the text (page 7, lines 254 and 255) it is mentioned that “PEGylated liposomes with the shortest (P800) and the longest (P2000) PEG-spacers were used.”
Corrected.
Page 8, lines 296 and 297: “After the injection of isRNA in complexes with conventional or PEGylated liposomes, TNF-α levels remained unchanged (Fig. 6B).” The size of the IL-6 bars masks the effect of isRNA-P1500 on TNF-α levels. Please consider presenting the TNF-α in a separate diagram from the IL-6. Please indicate statistical significance.
We evaluate statistical significance of the differences between IL-6 (white columns) and TNF-α (black columns) levels, shown in Fig.6, and revealed that differences in TNF-α levels are statistically insignificant. This is why we believe that presenting TNF-α in a separate diagram is not needed.
Minor points
Consider using dispersity index (Đ) instead of polydispersity index (PI)
Since in all our already published works we used polydispersity index (PI) to characterize liposomal formulation and lipoplexes in this study we also used PI instead of dispersity index (Đ) because it help to compare new results with already published.
Page 4, line 120: number instead of amount of living cells
Corrected
Page 10, line 319: Add reference where indicated.
Corrected
Page 1, line 43: “Cationic lipoplexes, which are a highly effective” delete the word a.
Corrected
Page 10, line 347: “…under the all conditions used.” delete the word the.
Corrected
Page 11, line 371: “…cause an immunostimulatory effect by alone” delete the word by.
Corrected
Page 13, line 482: lipoconjugate instead of lipoconjugat.
Corrected
Reviewer 2 Report
Good piece of work. The manuscript is interesting to read and the results reported in this work can provide some valuable information for designing nanocarriers for nucleic acid delivery. The manuscript can be improved according to the following comments:
In Figure 2, size, pdi, and zeta potential of formulation of NP=6/1 and 8/1 should be reported.
Apart from PEG length, PEG density is also an important factor affecting the performance of a nanocarrier. It is recommended that the authors should briefly mention in the Introduction or Discussion section, such as Biomaterials 2014, 35(9): 3027-3034; Biomaterials 2018, 182:72-81 et al.
Author Response
Dear Reviewer #2,
Thank you for careful study of our manuscript and for your very useful remarks and comments. We revised the manuscript and let us respond to your questions and comments.
REW2
Good piece of work. The manuscript is interesting to read and the results reported in this work can provide some valuable information for designing nanocarriers for nucleic acid delivery. The manuscript can be improved according to the following comments:
In Figure 2, size, pdi, and zeta potential of formulation of NP=6/1 and 8/1 should be reported.
To add size, pdi, and zeta potential of formulation of NP=6/1 and 8/1 to fig.2 will be difficult technically because the figure is rather busy. Than, we did not evaluate these parameters for those lipoplexes preparations shown in figure 2.
These characteristic were obtained for L/pDNA formulations (unpublished results); the data showed that at N/P ratios over 4/1 lipoplexes’ size PI and zeta potential remain similar:
L/pDNA - N/P 2:1 – 303 nm; PI – 0,257; zeta potential 42,4
N/P 4:1 - 195 nm; PI – 0,261; zeta potential 45,2
N/P 6:1 – 158 nm; PI - 0,248; zeta potential 48,3
N/P 8:1 – 161 nm; PI - 0,243; zeta potential 47,2
Apart from PEG length, PEG density is also an important factor affecting the performance of a nanocarrier. It is recommended that the authors should briefly mention in the Introduction or Discussion section, such as Biomaterials 2014, 35(9): 3027-3034; Biomaterials 2018, 182:72-81 et al.
Corrected, see page 10, line 328.
Round 2
Reviewer 2 Report
The authors have addressed my concerns.